# Pyramiding of bacterial blight resistance genes into promising restorer BRRI31R line through marker-assisted backcross breeding and evaluation of agro-morphological and physiochemical characteristics of developed resistant restorer lines

Anowara Akter[1]☯*, Lutful Hassan[2], Sheikh Arafat Islam Nihad[3]☯, Md. Jamil Hasan[1], Arif Hasan Khan Robin[2], Mahmuda Khatun[4], Anika Tabassum[5], Mohammad Abdul Latif[1]☯*

1 Hybrid Rice Division, Bangladesh Rice Research Institute (BRRI), Gazipur, Bangladesh, 2 Genetics and Plant Breeding, Bangladesh Agricultural University (BAU), Mymensingh, Bangladesh, 3 Plant Pathology Division, Bangladesh Rice Research Institute (BRRI), Gazipur, Bangladesh, 4 Genetics and Plant Breeding Division, Bangladesh Rice Research Institute (BRRI), Gazipur, Bangladesh, 5 Plant Breeding Division, Bangladesh Agricultural Research Institute (BARI), Gazipur, Bangladesh

☯ These authors contributed equally to this work.
* alatif1965@yahoo.com (MAL); anowaraa@yahoo.com (AA)

## Abstract

BRRI31R is one of the Bangladesh's most promising restorer lines due to its abundant pollen producing capacity, strong restoring ability, good combining ability, high outcrossing rate and genetically diverse from cytoplasmic male sterile (CMS) line. But the drawback of this line is that it is highly susceptible to bacterial blight (BB) disease of rice caused by *Xanthomonas oryzae* pv. *oryzae*. The present study highlighted the pyramiding of effective BB resistance genes (*xa5*, *xa13* and *Xa21*) into the background of BRRI31R, through marker–assisted backcrossing (MABC). Backcross progenies were confirmed and advanced based on the foreground selection of target genes. Pyramided lines were used for pathogenicity test against five Bangladeshi *Xanthomonas oryzae* (*BXo*) races (*BXo*93, *BXo*220, *BXo*822, *BXo*826, *BXo*887) and confirmed the dominant fertility restore genes, *Rf3* and *Rf4* and further validated against SNP markers for more confirmation of target resistance genes. All pyramided restorer lines consisted of *Xa4* (in built), *xa5*, *xa13*, *Xa21*, and *Chalk5* with two fertility restorer genes, *Rf3*, *Rf4*. and these restorer lines showed intermediate amylose content (<25%). Restorer lines BRRI31R-MASP3 and BRRI31R-MASP4 showed high levels of resistance against five virulent *BXo* races and SNP genotyping revealed that these lines also contained a blast resistance gene *Pita* races. Gene pyramided restorer lines, BRRI31R-MASP3 and BRRI31R-MASP4 can directly be used as a male parent for the development of new BB resistant hybrid rice variety or could be used as a replacement of restorer line of BRRI hybrid dhan5 and 7 to enhance the quality of hybrid seeds as well as rice production in Bangladesh.

**Data Availability Statement:** All relevant data are within the manuscript and its Supporting Information files.

**Funding:** Authors received funds from Strengthening Physical Infrastructure and Research Activities (SPIRA) Project, Bangladesh Rice Research Institute, through the Ministry of Agriculture, Bangladesh to conduct this experiment. The funders had no role in study design, data collection and analysis, decision to publish, or preparation of the manuscript.

**Competing interests:** The authors have declared that no competing interests exist.

# Introduction

Rice (*Oryza sativa* L.) is the world's most influential and economically significant crop, with more than half of the world's population relying on rice to meet their daily calorie requirements [1, 2]. Bangladesh has secured the third position for total rice production in the world and in the country food security depends on rice security [1, 2]. It has been assumed that, in 2050, the population of Bangladesh will be 215.4 million, when 44.6 MT of clean rice will be required [3]. Climate change and the emergence of different biotic and abiotic stresses are threatening rice production in the world including Bangladesh [1, 4, 5]. To ensure the basic food demand of the burgeoning population of Bangladesh, hybrid rice could be a viable solution to increase rice production with a 15–20% yield advantage compared to inbred varieties [6, 7].

In Bangladesh, till now, 32 rice diseases have been reported and based on severity and incidence, blast [8, 9], sheath blight [10], BB [11, 12], tungro [13] and false smut [14] considered as major diseases across the country. At present, BB is one of the major threats to both inbred and hybrid rice cultivation in the world specifically in Southeast Asia including Bangladesh [15–19]. In South and Southeast Asian nations, BB causes 10–50% yield loss, but it can exceed 80%, depending on variety, crop stage, location, and favorable environment [20–22]. The input requirement of hybrid is more compared to inbred varieties and these phenomena favor the epidemics of diverse diseases and pests. High-yielding varieties require high dose of nitrogen which favors the occurrence of BB in different countries of the world including Bangladesh [22]. Ansari et al. [22] found that due to BB, around 5.8–30.4% yield loss occurred in field conditions of Bangladesh. However, the damage caused by bacterial blight disease depends on many factors including location, virulence of the race, crop stage, environmental condition and cultivar [23, 24]. But till now, no effective chemical control measure is suggested to combat the disease. So far, 237 hybrid rice varieties have been released in Bangladesh but no variety has any resistance characteristics because all hybrid parental lines are susceptible to available races of *Xoo* [25].

Therefore, stacking resistance genes into hybrid parental lines is a durable strategy to manage bacterial blight disease [25]. Moreover, by adopting conventional breeding it is tough to introgress more than one genes into the single lines and sometimes dominance and epistasis effects make it more difficult to identify the effective gene (s) [26, 27]. However, the development of DNA markers linked to resistance genes makes it easy to detect two or more genes in the introgressed lines [8, 27–29]. Trait-specific gene transformation through marker-assisted selection (MAS) is a very popular technique to develop inbred and hybrid lines with desirable characters [8, 30]. To date, more than 46 resistance genes have been identified as effective against bacterial blight pathogens [31]. Among them, *Xa4*, *xa5*, *Xa7*, *xa13*, *Xa21*, and *Xa23* genes have been extensively deployed and incorporated into rice cultivars for the development of BB resistant cultivars [32]. Resistance genes i.e., *xa5*, *xa13* and *Xa21* were found effective for the bacterial blight pathogens of Bangladesh [23, 25, 33, 34]. Moreover, molecular markers of these resistance genes are widely used in breeding programs to develop bacterial blight resistant rice varieties [9, 35, 36]. It is crucial to characterize genotypes using trait-specific SNP markers in order to examine the frequency of advantageous alleles for various traits of interest associated with biotic and abiotic stressors as well as grain quality characteristics. The high throughput SNP platform has made it possible to choose genotypes for elite parents with exceptional traits utilizing contemporary breeding techniques like marker-assisted forward breeding and genomic selection.

As a result of the aforementioned facts, the current study was initiated with the following goals in mind: (i) introgression of *xa5*, *xa13* & *Xa21* genes in the promising restorer lines,

BRRI31R using MABC; (ii) assessment of introgressed pyramided restorer lines through molecular approach and pathogenicity test against bacterial blight pathogen(iii) confirmation of fertility restorer genes (iv) evaluation of morphological and grain quality characters of developed pyramided lines in comparison with respective recurrent parent. However, the development of pyramided BB resistant restorer lines carrying four resistance genes (*Xa4*, *xa5*, *xa13* and *Xa21*) have a promising future to develop a bacterial blight-resistant hybrid variety in Bangladesh.

## Materials and methods

### Plant materials

BRRI31R is a restorer background with medium long slender, medium duration, strong restorers but susceptible to bacterial blight disease. IRBB60, a pyramided line carrying four resistance genes (*Xa4* + *xa5* + *xa13*+ *Xa21*) was used as the donor for introgression of BB resistance genes into the most promising restorer line BRRI31R. We considered this line as promising because this parental line was also used in the background of two popular hybrid rice varieties (BRRI hybrid dhan5 and BRRI hybrid dhan7) of Bangladesh. Before hybridization, the selected parents were used for phenotyping screening against five *BXo* races and validation of markers against BB resistance genes, *Xa4*, *xa5*, *xa13* and *Xa21* for foreground selection.

### Marker-assisted breeding strategy for restorer line development

A cross was made between BRRI31R and IRBB60. The $F_1$ plants were backcrossed thrice with recurrent parent (BRRI31R) to produce $BC_3F_1$ plants, which were then selfed to get up to $BC_3F_5$ progenies. Whole backcross was confirmed and advanced from $BC_3F_2$ to $BC_3F_5$, based on gene-linked molecular markers. $F_1$ to $BC_3F_1$ were followed for background recovery of recurrent parent and foreground selection was started from$F_1$ generation and continued till $BC_3F_5$ (Fig 1) where pure homozygous lines for all three resistance genes were detected. The selected and confirmed pyramided restorer lines were used for pathogenicity test against five *BXo* races.

### DNA extraction and PCR amplification

Leaves of twenty-one days old seedlings were obtained from the field to extract DNA. The adapted CTAB (Cetyl Trimethyl Ammonium Bromide) protocol described by Nihad et al., [37] was used for DNA extraction. DNA quality was measured by a nanodrop spectrophotometer (Nanodrop$^{TM}$ 2000/2000c) and kept at –20˚C for further use. Four linked/gene-specific molecular markers, MP1-F & MP2-R, *xa5*_F2_Sus. Forward + *xa5*_F2_res. Forward + *xa5*_R2 reverse primer (common), xa13F_130_147 + xa13R_1678_1662 and pTA248 linked to *Xa4*, *xa5*, *xa13* and *Xa21*, respectively, were used for foreground selection in each backcross and selfing population. PCR components, 3 μL DNA, 1 μL primer (0.5 μL forward and 0.5 μL reverse primers), 5 μL GoTaq® G2 Green Master Mix (2X) (Promega, USA) and 1.0 μL double distilled (nuclease-free) water were taken in 0.2 mL PCR tube to conduct the PCR reaction. Details of gene-linked/specific primers and their PCR programs are given in Table 1 and S1 Table. To visualize the DNA band, electrophoresis was done on 1.5% agarose gel followed by ethidium bromide staining.

### Validation of pyramided restorer lines by SNP markers

Finally, selected pyramided restorer lines were validated by SNP markers against 20 QTL linked to different traits which are shown in the S2 Table.

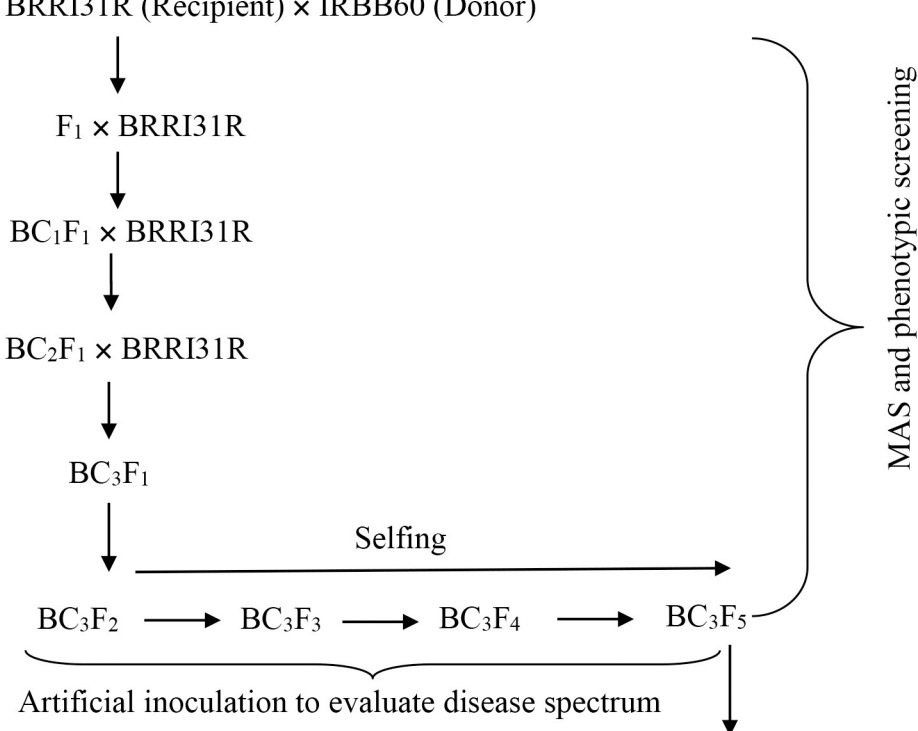

Fig 1. Schematic depiction of the introgression of *xa5*, *xa13* and *Xa21* into the restorer line.

## Molecular study for fertility status of pyramided restorer lines

Selected five pyramided restorer lines were tested for fertility restorer genes, *Rf3* and *Rf4* by using molecular markers (Table 1). A fertility restorer and non–fertility restorer lines were used as a check to confirm the presence of fertility restorer genes, *Rf3* and *Rf4* in the pyramided restorers lines during Transplanted Aman season, 2020 (wet season).

## Conventional approaches to study fertility status of pyramided restorer lines

**Constitute of source nursery.** A total of five pyramided restorer lines along with two wilds abortive (WA) CMS lines, IR75608A and IR79156A were constituted in the source nursery during the Boro season (dry season), 2020. Each line was grown in three (3) sets with different planting times for proper synchronization of flowering to make a test cross for the validation of fertility status.

**Table 1. Molecular markers used for foreground selection.**

| Gene | Chr. | Marker | Sequences (5'-3') | Expected allele size (bp) | Reference |
|---|---|---|---|---|---|
| *Xa4* | 11 | MP1 | F–ATCGATCGATCTTCACGAGG | 150 | Suh et al., [38] |
| | | MP2 | R–TGCTATAAAAGGCATTCGGG | | |
| *xa5* | 5 | *xa5*_F2 | SF–GCTCGCCATTCAAGTTCTTGTC | 198 | IRRI [39] |
| | | *xa5*_F2 | RF–GCTCGCCATTCAAGTTCTTGAG | | |
| | | *xa5*_R2 (Common) | RP–CCTTGATAGAAACCTTGCTCTTGAC | | |
| *xa13* | 8 | *xa13F*_130_147 | F–CCTGATATGTGAGGTAGT | 1520 | Kumar et al., [40] |
| | | *xa13R*_1678_1662 | R–GAGAAAGGCTTAAGTGC | | |
| *Xa21* | 11 | pTA248 | F–AGACGCGGAAGGGTGGTTCCCGGA | 1000 | Xu et al., [41] |
| | | | R–AGACGCGGTAATCGAAAGATGAAA | | |
| *Rf3* | 1 | RM1 | F–GCGAAAACACAATGCAAAAA | 115 | Revathi et al., [42] |
| | | | R–GCGTTGGTTGGACCTGAC | | |
| *Rf4* | 10 | DRCG-RF4-14F &14R | F–GCAATGCTTGTATTCAGCAAA | 800 | Suresh et al., [43] |
| | | | R–TCCAGCTGTAAATCCGTCAA | | |

Note. F:Forward, R:Reverse, Chr.:Chromosome.

**Evaluation of test cross $F_1$s.** A total of ten (10) test cross $F_1$s (S3 Table) were evaluated in test cross nursery by two ways such as a microscopic study and a phenotypic study for the confirmation of fertility status at the flowering stage by following the scale of Hybrid Rice Breeding Manual, IRRI, Philippines 1997 i.e., Full fertile (FF): 81–100% fertility, Fertile: 61–80%, Partially fertile (PF): 31–60%, Partially sterile (PS): 21–20%, Sterile (S): 1–20% and Completely sterile (CS): 0% fertility.

**Pollen fertility and spikelet fertility status of testcross $F_1$s.** The pollen grains were put on a glass slide and dyed with one drop of 1% Iodine Potassium Iodide (IKI) solution. The anthers were gently smashed with a needle to release pollen grains, and pollen fertility was determined using a compound microscope at 10X magnification. Spikelet fertility percentage (seed sett %) was counted manually. When the dark black cell was seen in IKI solution under a microscope, it meant that male parents were confirmed to be suspected restorers. The fertility of pollen and spikelet was measured by the following formula: Pollen fertility (%) = (No. of fertile pollen/No. of total pollen)*100; Spikelet fertility (%) = (Filled spikelets per panicle/Total spikelets per panicle)*100.

**Screening for BB resistance.** The parent and selected pyramided restorer lines were inoculated with five bacterial blight races (*BXo93, BXo220, BXo826, BXo822 and BXo887*) suspension ($OD_{600} = 1$ equivalent to bacterial cell concentration $3.3 \times 10^8$ CFU/mL) following the adapted clipping procedure of Rashid et al., [23] modified from Kauffman et al., [44]. At the maximum tillering stage, ten hills per entry were selected for *BXo* race inoculation and 8–10 leaves were trimmed from the apex. Before plunging into the fresh isolate mixture, the scissors were rinsed three times with sterilized water and then sterilized in 70% ethanol for 30 minutes. Following 14 days of inoculation, eight (8) infected leaves per hill were collected from the field for lesion length measurement. The disease response was graded using the modified scale of Akter et al., [25]. Briefly, disease data was categorized into six scales (0 to 9) i.e., 0-Highly resistant (HR, lesion length <0.5 cm), 1-Resistant (R, lesion length 0.5–3.0 cm), 3-Moderately resistant (MR, lesion length 3.1–5.0 cm), 5-Moderately susceptible (MS, lesion length 5.1–10.0 cm), 7-Susceptible (S, lesion length 10.1–15.0 cm) and 9-Highly susceptible (HS, lesion length >15.0 cm).

**Agro-morphological traits and grain quality test of pyramided restorer lines.** Five pyramided restorer lines and their recurrent parent were seeded in the Transplanted Aman season

(wet season) of 2020. Twenty-one days old seedling was transplanted with a spacing of 20 cm × 15 cm and other agronomic practices. Data was recoded on five plants from each pyramided restorer line in BC$_3$F$_5$ generation for agronomic traits like plant height (cm), days to 50% flowering (day), days to maturity (day), number of effective tiller/hill, panicle length (cm), number of spikelet/panicle, spikelet fertility (%), 1000 grain weight (g), Grain yield/ hills (g). On the other hand, grain quality parameters such as milling outturn, head rice recovery, milled rice length, milled rice breadth, length breadth ratio, amylose content, protein content, cooking time, elongation ratio and imbibition ratio were measured in the laboratory of Grain Quality and Nutrition Division of Bangladesh Rice Research Institute (BRRI), Gazipur-1701, during 2020.

**Statistical analysis of the data.** Replication means data of the yield contributing traits and physicochemical and cooking characters under the study were subjected to univariate analysis. Univariate analysis of the individual character was done by computer using Excel program 2010 software. Statistical analysis was performed with independent samples using least significance difference (LSD) at 5% level and coefficient of variation percentage (CV%). Selected five pyramid restorer lines were genotyped with 20 trait-based single nucleotide polymorphism (SNP) markers developed by the International Rice Research Institute (IRRI; https://www.irri.org/; https://gsl.irri.org/) [45] utilizing Kompetitive allele-specific PCR (KASP) assay for high-precision bi-allelic characterization of SNP with Intertek (https://www. intertek.com/agriculture/agritech/) as a service provider. The SNP markers associated with the trait of interests such as snpOS00493, snpOS00061, snpOS00054 for BB resistant; snpOS00478, snpOS00451, snpOS00006, snpOS00468 for blast resistant; snpOS00430 and snpOS00442 for brown planthopper; snpOS00466, snpOS00467 for gall midge; snpOS00445, snpOS00446, snpOS00038 for amylose content; snpOS00024 for chalkiness (grain quality); snpOS00396 for grain number and snpOS00409, snpOS00410, snpOS00411, snpOS00397 for salt were assayed.

## Results

### Validation of markers for foreground selection of parents

The repeatability and polymorphism of gene-based/linked markers were checked between the recurrent parent, BRRI31R and donor parent IRBB60 (Table 2). Among the parents, the restorer line BRRI31R showed monomorphic nature with a resistant allele (150bp) of the *Xa4* gene but showed susceptible alleles (198bp, 1322bp and 650bp) for *xa5*, *xa13* and *Xa21* genes with polymorphic nature, respectively. Gel photographs of the markers for the four bacterial blight resistance genes in the parents are presented in Fig 2A.

### Pathogenicity test of parents against bacterial blight pathogen

After the morphological screening, the recipient parent, BRRI31R showed (23.5 cm) similar mean lesion length compared to susceptible check IR24 (23.8 cm) while the resistant check

**Table 2. Evaluation of parents by BB resistance genes *Xa4*, *xa5*, *xa13* and *Xa21* using MAS and disease reaction.**

| SN. | Parents | BB resistance genes | | | | Isolates/Lesion length (cm) | | | | | Mean lesion length (cm) | Disease reaction |
|---|---|---|---|---|---|---|---|---|---|---|---|---|
| | | *Xa4* | *xa5* | *xa13* | *Xa21* | *BXo 93* | *BXo 220* | *BXo 822* | *BXo 826* | *BXo 887* | | |
| 1 | BRRI31R (RP) | P | A | A | A | 21.7 | 24.1 | 23.2 | 25.1 | 23.6 | 23.5 | HS |
| 2 | IRBB60 (DP & RC) | P | P | P | P | 1.0 | 0.8 | 0.8 | 0.9 | 0.8 | 0.9 | R |
| 3 | IR24 (SC) | A | A | A | A | 20.9 | 22.7 | 25.1 | 25.0 | 25.6 | 23.8 | HS |

Note. RP:Recurrent parent, DP:Donor parent, RC:Resistant check, SC:Susceptible check, HS:Highly susceptible, R:Resistant, SN:Serial number.

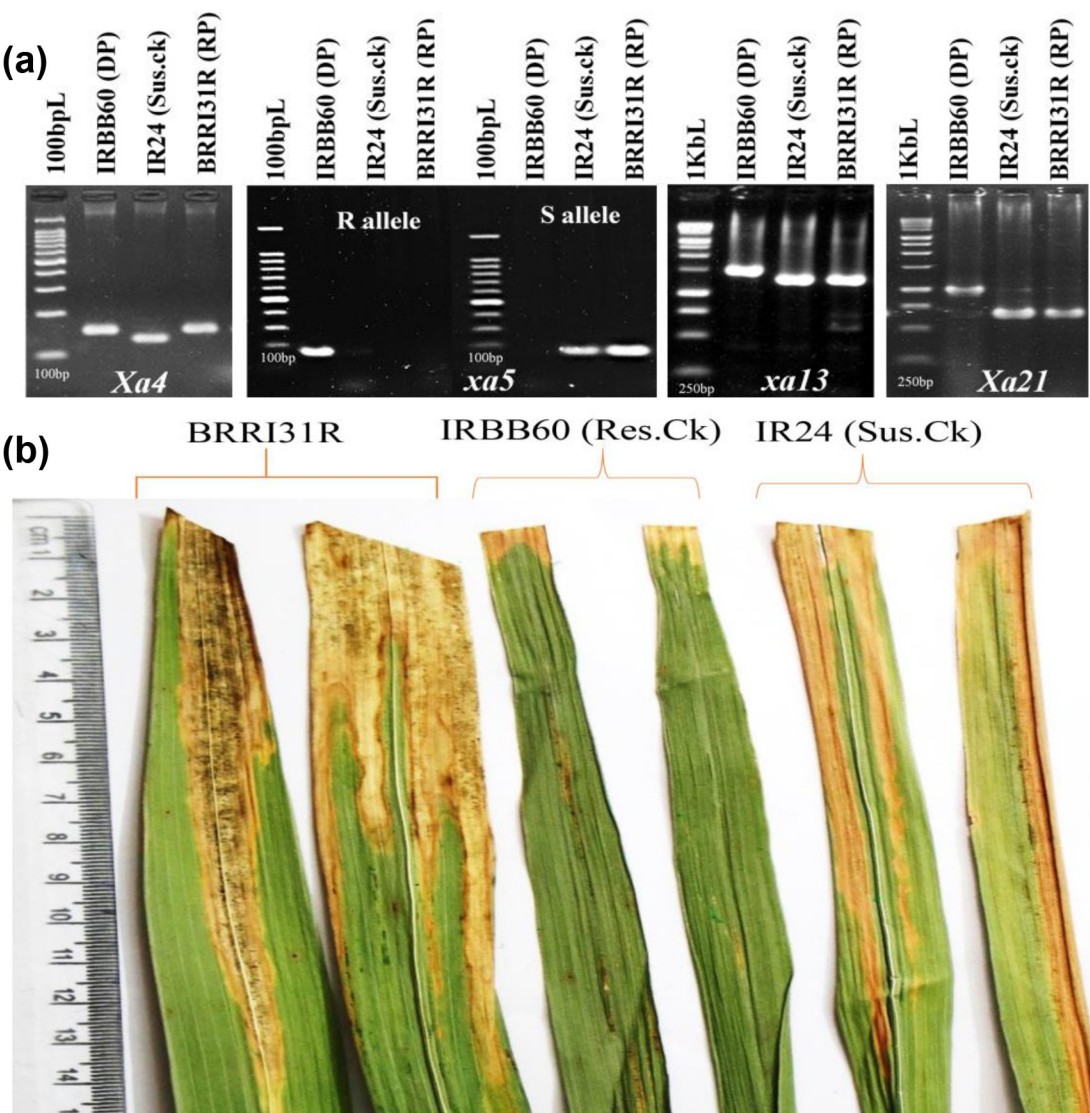

**Fig 2.** (a) Gel photographs showing banding pattern of the markers for the four bacterial blight resistance genes (*Xa4*, *xa5*, *xa13* and *Xa21*) in the parents (b) Phenotypic screening of selected parents along with resistant and susceptible check against *BXo* races of bacterial blight pathogen.

showed mean lesion length of 0.9 cm (Table 2). The disease reaction of parents along with susceptible check against bacterial blight pathogen is shown in Fig 2B.

## Stacking of bacterial blight resistance genes into BRRI31R using MABC

Fifty heterozygous $F_1$ plants obtained from the cross between BRRI31R × IRBB60 were confirmed through molecular markers. Out of fifty $F_1$ plants, ten heterozygous plants with three BB resistance genes (*xa5*, *xa13* and *Xa21*) were backcrossed to respective recurrent parent (BRRI31R). In $BC_1F_1$, foreground selection with target genes resulted in the selection of heterozygous plants. A total of 24 out of 55 heterozygous plants were found with triple BB resistance genes and these plants were backcrossed with a recurrent parent to obtain $BC_2F_1$ population. In $BC_2F_1$, a total of 22 out of 62 plants were found with three resistant alleles in heterozygous condition for *xa5*, *xa13* and *Xa21* genes and these plants were further

**Table 3. Marker analysis in segregating BC$_3$F$_2$ population for $xa5$, $xa13$ and $Xa21$ genes.**

| Genes/Markers | Observed value | | Expected ratio | Expected value | $\chi^2$-value | Probability |
|---|---|---|---|---|---|---|
| **BRRI31R × IRBB60** | | | | | | |
| $xa5$ | RR = R | 45 | 1 | 50 | 0.500 | 0.232 |
| | Rr = H | 112 | 2 | 100 | 1.440 | |
| | rr = S | 43 | 1 | 50 | 0.980 | |
| **Total** | | 200 | 4 | 200 | **2.920** | |
| $xa13$ | RR = R | 52 | 1 | 50 | 0.080 | 0.423 |
| | Rr = H | 106 | 2 | 100 | 0.360 | |
| | rr = S | 42 | 1 | 50 | 1.280 | |
| **Total** | | 200 | 4 | 200 | **1.720** | |
| $Xa21$ | RR = R | 49 | 1 | 50 | 0.020 | 0.348 |
| | Rr = H | 109 | 2 | 100 | 0.810 | |
| | rr = S | 42 | 1 | 50 | 1.280 | |
| **Total** | | 200 | 4 | 200 | **2.110** | |

Note. R:Resistant, S:Susceptible, H:Heterozygous, RR:Plants with a banding pattern alike to the resistant parent alleles, Rr:Heterozygous plants, rr:Plants with a banding pattern alike to the susceptible parent alleles, $\chi^2$: Actual value of the chi-square test for resistant/susceptible ratio.

backcrossed to produce BC$_3$F$_1$ population. From BC$_3$F$_1$ population, 24 out of 65 plants were selected with three BB resistance genes and selfed to produce BC$_3$F$_2$ progenies. Finally, two hundred BC$_3$F$_2$ progenies were tested using linked markers to identify homozygous plants for different $R$ genes of BRRI31R × IRBB60. Plants with single, double and triple gene combinations with BB resistance were identified in this study. Among them, 45 plants were shown with homozygous resistance for $xa5$, 52 for $xa13$ and 49 for $Xa21$, respectively. Simultaneously, 23 plants for $xa5$ + $xa13$, 25 plants for $xa5$ + $Xa21$, 17 for $xa13$ + $Xa21$ and 15 for $xa5$ + $xa13$ + $Xa21$ genes were found as homozygous. One hundred and seventy plants were found homozygous to susceptible alleles like as recurrent parent. Finally, five (5) homozygous plants for three resistance genes ($xa5$, $xa13$ and $Xa21$) were selected based on molecular screening and phenotypically compared to respective recurrent parents. The segregation of bacterial blight resistance in BC$_3$F$_2$ population was analyzed using the chi-square ($\chi2$) test. The $\chi^2$ values value of segregation analysis for $xa5$, $xa13$ & $Xa21$ genes was insignificant as the calculated $\chi^2$ values, 2.920 ($xa5$), 1.720 ($xa13$) and 2.110 ($Xa21$) is lesser than the tabulated values at 5% (5.991) and 1% (9.210) levels (Table 3). The selected five homozygous BC$_3$F$_2$ plants were forwarded to BC$_3$F$_3$ and continued from BC$_3$F$_3$ to BC$_3$F$_5$ progenies. BRRI31R had by default $Xa4$ gene (Fig 2A) and for this reason, further confirm the four BB resistance genes ($Xa4$, $xa5$, $xa13$ and $Xa21$) through gene-based/linked markers in the cross combination, BRRI31R × IRBB60. Five pyramided restorer lines with four BB resistance genes were confirmed through molecular markers (Fig 3A and 3D). In addition, these lines were used in the evaluation nursery for pathological screening against five $BXo$ races and for the confirmation of fertility status. Validation of BB resistance genes with additional traits were also done by using 20 SNP panels and Moreover, agro-morphological and grain quality were also assessed to check the phenotypic and quality performance of the restorer lines in comparison with the recurrent parent.

## Screening of selected pyramided lines against bacterial blight pathogen

The BB resistance of pyramided restorer lines with the parents and susceptible check IR24 were evaluated by five $BXo$ races. The mean lesion length of pyramided restorers ($Xa4$, $xa5$, $xa13$ and $Xa21$) was very less (<3 cm) compared to recurrent parents and susceptible check

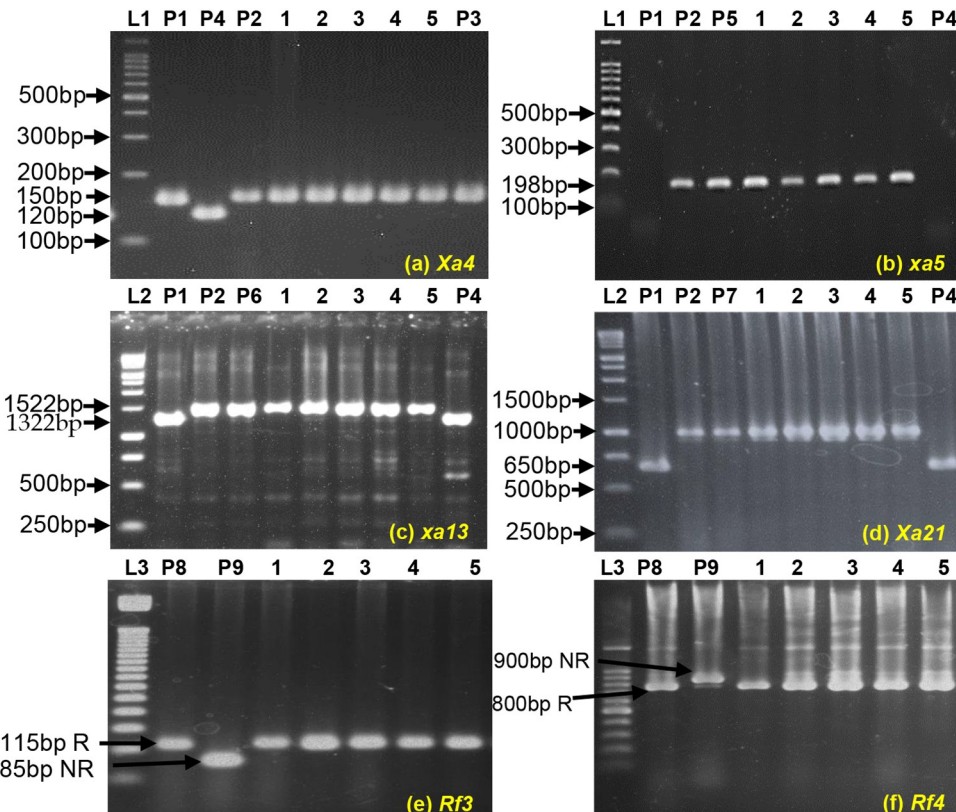

**Fig 3.** Gel image of *Xa4* (a), *xa5* (b), *xa13* (c), *Xa21* (d), *Rf3* (e) and *Rf4* (f) genes in fixed restorer lines of $BC_3F_5$ generation. Here, L1:100bp, L2:1kb, L3:50bp, P1:BRRI31R (recipient parent), P2:IRBB60 (donor parent), P3:IRBB4 (resistant check for *Xa4* gene), P4:IR24 (susceptible check), P5:IRBB5 (resistant check for *xa5* gene), P6:IRBB13 (resistant check for *xa13* gene), P7:IRBB21 (resistant check for *Xa21* gene), P8:IR96479-81-7-1-1-B-1-1-1-R (RA, restorer allele), P9:IR58025A (NR, non-restorer allele).

IR24 (Fig 4). Out of five, two pyramided restorer lines, BRRI31R-MASP3 and BRRI31R--MASP4 were showed highly resistant reaction against all the races (*BXo93, BXo220, BXo826, BXo822 and BXo887)* with mean lesion length <0.5 cm, which was similar to the mean lesion of the donor parent IRBB60. This result indicated that the introgression of *xa5, xa13* and *Xa21* genes significantly enhanced the resistance of the improved pyramided restorer lines to BB. The mean lesion length (cm) of all pyramided restorer lines (resistant to highly resistant) with parents and susceptible check are shown in Table 4.

### Evaluation of fertility status of pyramided restorer lines

**Observation of fertility status by gene-based/linked markers.** Five pyramid restorer lines were subjected to PCR analysis using gene-based or linked markers against fertility restore genes *Rf3* & *Rf4*. All pyramid restorer lines showed restorer alleles, *Rf3* and *Rf4* where band size was 115bp and 800bp, respectively (Fig 3E and 3F). So, these lines had the ability to restore fertility.

**Observation of fertility status in testcross $F_1$s by the conventional way.** Out of ten, four testcrosses of $F_1$s combinations, IR75608A × BRRI31R-MASP3, IR75608A × BRRI31R-MASP4, IR79156A × BRRI31R-MASP3 and IR79156A × BRRI31R-MASP4 were found having the highest pollen fertility (90.45%, 91.75%, 94.15% and 92.89%, respectively) and seed sett (94.64%, 93.88%, 95.10% and 94.46%,

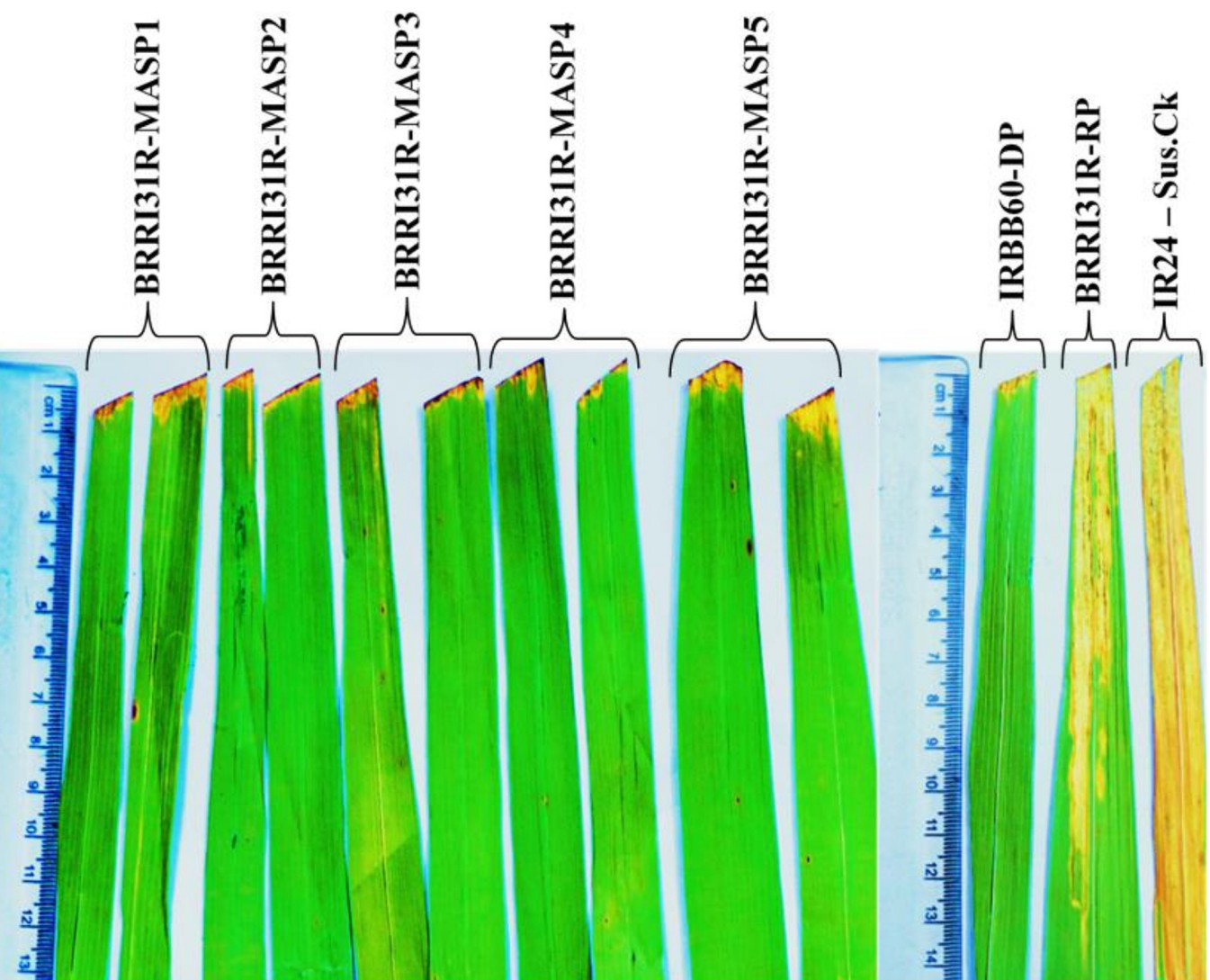

**Fig 4. Disease reaction of selected pyramided restorer lines along with respective recurrent parent, donor parent and susceptible check against *BXo* races of bacterial blight pathogen.**

respectively) percentage (Table 5). $F_1$ seed sett or spikelet fertility of these testcross above 90% indicating that these were pyramided restorer lines (BRRI31R-MASP3 and BRRI31R-MASP4) denoted strong fertility status with restorer genes *Rf3* and *Rf4* against IR75608A and IR79156A.

## Pyramided restorer lines validated by SNP markers

Selected five pyramid restorer lines were validated by SNP markers linked with 20 QTL of rice traits. Two pyramided restorer lines, BRRI31R-MASP3 and BRRI31R-MASP4 consisted of target QTL *xa5* (resistant allele AG:AG by snpOS00054), *xa13* (resistant allele C:C by snpOS000493), *Xa21* (resistant allele C:C by snpOS00061), *Pita* (resistant allele C:C by snpOS00006), *Chalk5* (resistant allele G:G by snpOS00024) but another three pyramid restorer lines, BRRI31R-MASP1, BRRI31R-MASP2 and BRRI31R-MASP5 contained all of the above

**Table 4. Evaluation of pyramided restorer lines and parents against bacterial blight pathogen.**

| Pyramided restorer lines/parents/Sus.ck | Lesion length (cm) | | | | | Mean lesion length (cm) | Disease reaction |
|---|---|---|---|---|---|---|---|
| | *BXo93* | *BXo220* | *BXo822* | *BXo826* | *BXo887* | | |
| **BRRI31R-MASP1** | 1.0 | 1.1 | 1.5 | 1.2 | 1.6 | 1.3 | R |
| **BRRI31R-MASP2** | 1.2 | 1.5 | 1.4 | 1.1 | 1.0 | 1.2 | R |
| **BRRI31R-MASP3** | 0.2 | 0.3 | 0.4 | 0.1 | 0.5 | 0.3 | HR |
| **BRRI31R-MASP4** | 0.1 | 0.2 | 0.3 | 0.4 | 0.2 | 0.2 | HR |
| **BRRI31R-MASP5** | 1.1 | 0.9 | 1.4 | 1.3 | 1.5 | 1.2 | R |
| **IRBB60-DP** | 0.2 | 0.3 | 0.4 | 0.1 | 0.5 | 0.3 | HR |
| **BRRI31R-RP** | 22.7 | 24.5 | 23.8 | 25.5 | 24.2 | 24.1 | HS |
| **IR24—Sus.ck** | 24.7 | 25.6 | 26.1 | 25.2 | 26.5 | 25.6 | HS |

Note. DP:Donor parent, RP:Recurrent parent, Sus.ck:Susceptible check, R:Resistant, HR:Highly resistant, HS:Highly susceptible.

genes without blast resistance gene *Pita* (resistant allele C:C by snpOS00006) (Table 6). After pathogenicity test and validation with molecular and morphological study, out of five, two pyramided restorer lines, BRRI31R-MASP3 and BRRI31R-MASP4 were selected based on the best phenotypic similarity to the recurrent parent with blast resistance gene, *Pita*. For the sake of clarification, disease reaction of the *Pita* gene was not tested here.

## Agro-morphological and grain quality test of pyramided restorer lines

No significant differences were found between the pyramided restorer lines and the respective recurrent parent. All the selected pyramided restorer lines had alike agro-morphological and grain-quality characters as recipient parent (Tables 7 and 8 and Fig 5). Out of the five two pyramided restorer lines, BRRI31R-MASP3 and BRRI31R-MASP4 produced higher yields than the recipient parent.

## Discussion

BB disease is a key limitation for hybrid rice farming in Bangladesh since most hybrid rice parental lines are susceptible to BB. Introgression of bacterial blight resistance genes into parental lines of hybrid rice using MABC is the best way to develop durable bacterial blight resistant lines. Because through MABC, the desired genes can be easily transferred into the

**Table 5. Restoration confirmation of pyramided restorer lines.**

| SN. | Testcross F$_1$s of pyramided lines | Pollen fertility (%) | Spikelet fertility or seed set % | Category | Remarks |
|---|---|---|---|---|---|
| **1** | IR75608Ax BRRI31R-MASP1 | 87.20 | 89.80 | FF | Restorer |
| **2** | IR75608Ax BRRI31R-MASP2 | 86.00 | 87.20 | FF | Restorer |
| **3** | IR75608Ax BRRI31R-MASP3 | 90.45 | 94.64 | FF | Restorer |
| **4** | IR75608Bx BRRI31R-MASP4 | 91.75 | 93.88 | FF | Restorer |
| **5** | IR75608B x BRRI31R-MASP5 | 88.22 | 88.35 | FF | Restorer |
| **6** | IR79156Ax BRRI31R-MASP1 | 89.22 | 91.78 | FF | Restorer |
| **7** | IR79156Ax BRRI31R-MASP2 | 91.95 | 92.50 | FF | Restorer |
| **8** | IR79156Ax BRRI31R-MASP3 | 94.15 | 95.10 | FF | Restorer |
| **9** | IR79156Ax BRRI31R-MASP4 | 92.89 | 94.46 | FF | Restorer |
| **10** | IR79156Ax BRRI31R-MASP5 | 93.33 | 91.99 | FF | Restorer |

Note. FF:Full Fertile, SN:Serial number.

**Table 6. List of pyramided restorer lines and parents along with important traits revealed by SNP genotyping.**

| SN. | Pyramided restorers lines/parents | Important traits present |
|---|---|---|
| 1 | BRRI31R-MASP1 | *Xa4, xa5, xa13, Xa21, Chalk5, Rf3, Rf4* |
| 2 | BRRI31R-MASP2 | *Xa4, xa5, xa13, Xa21, Chalk5, Rf3, Rf4* |
| 3 | BRRI31R-MASP3 | *Xa4, xa5, xa13, Xa21, Pita, Chalk5, Rf3, Rf4* |
| 4 | BRRI31R-MASP4 | *Xa4, xa5, xa13, Xa21, Pita, Chalk5, Rf3, Rf4* |
| 5 | BRRI31R-MASP5 | *Xa4, xa5, xa13, Xa21, Chalk5, Rf3, Rf4* |
| 6 | BRRI31R-RP | *Xa4, Chalk5, Pita, Rf3, Rf4* |
| 7 | IRBB60-DP | *Xa4, xa5, xa13, Xa21, Chalk5* |

Note. RP:Recurrent Parent, DP: Donor Parent, SN:Serial number.

elite lines and can quickly recover the recurrent parent characteristics with a minimum number of backcrosses. We developed five pyramided lines having *Xa4*, *xa5*, *xa13* and *Xa21* genes with fertility restorer genes. Many reviewers previously studied and introgressed BB resistance genes into the promising restorer lines for the development of BB resistant pyramid hybrid rice [46–49]. MAS offers a simpler and more efficient method to improve rice cultivars or lines and marker-assisted introgression of major genes/QTLs has helped to develop resistant varieties against major diseases such as BB, blast, tungro etc. [8, 29, 50–52]. But conventional backcross cannot precisely transfer more than one gene into the cultivars and also the process needs a significant amount of time [38, 53, 54]. In this study, selected parents, BRRI31R and IRBB60 were validated against four genes–based or genes-linked markers for foreground selection. It was observed that the restorer line, BRRI31R showed monomorphic nature with a resistant allele against the *Xa4* gene but showed a susceptible allele against the *xa5*, *xa13* and *Xa21* genes.

In this study, BC$_3$F$_5$ pyramid lines have been developed by backcrossing and selfing, and at each stage, BB resistant plants were selected based on molecular screening by MAS, pathogenicity test and phenotype comparison to their respective recurrent parent. The similar approach followed by many researchers and successfully introgressed two or three or four BB resistance genes into the rice cultivars or lines [28, 38, 55, 56]. Our restorer lines were selected with target BB resistance genes using molecular markers and so there is a chance to carry linkage drag with the target gene. Backcross breeding is helpful to remove the linkage drug [57, 58] and to eliminate linkage drag, backcross progenies were selected in each backcross and

**Table 7. Agro-morphological traits of pyramided restorer lines and recurrent parent.**

| Pyramided lines/recurrent parent | PH (cm) | DFF | DTM | ET/P | PL (cm) | G/P | SF (%) | TGW (g) | GY/P(g) |
|---|---|---|---|---|---|---|---|---|---|
| **BRRI31R-MASP1** | 115.2 | 107 | 132 | 9 | 28.3 | 215 | 91 | 31.8 | 38.8 |
| **BRRI31R-MASP2** | 114.9 | 108 | 133 | 10 | 28.4 | 212 | 93 | 31.5 | 39.5 |
| **BRRI31R-MASP3** | 115.0 | 105 | 130 | 12 | 28.8 | 225 | 95 | 32.0 | 42.2 |
| **BRRI31R-MASP4** | 115.3 | 102 | 127 | 11 | 28.7 | 220 | 94 | 31.9 | 41.5 |
| **BRRI31R-MASP5** | 115.5 | 106 | 131 | 10 | 28.5 | 218 | 90 | 31.7 | 40 |
| **BRRI31R-RP** | 115.1 | 109 | 134 | 10 | 28.5 | 208 | 93 | 32.2 | 40.2 |
| **Mean** | 115.2 | 106.2 | 131.2 | 10.3 | 28.5 | 216.3 | 92.7 | 31.9 | 40.4 |
| **LSD (5%)** | 0.21 | 2.39 | 2.39 | 0.99 | 0.18 | 5.79 | 1.79 | 0.23 | 1.22 |
| **CV (%)** | 0.17 | 2.14 | 1.73 | 9.12 | 0.60 | 2.54 | 1.83 | 0.70 | 2.86 |

Note. PH:Plant height, DFF:Days to 50% flowering, DTM:Days to maturity, ET/P:Effective tiller/Plant, PL:Panicle length (cm), G/P:Grains per panicle, SF:Spikelet fertility in percent, TGW:Thousand grain weight (g), GY/P:Grain yield per plant (g), LSD:Least significant difference at 5% probability level, CV:Coefficient of variation.

**Table 8. Mean performance of grain quality parameters of pyramided restorer lines and recurrent parent.**

| Pyramided lines/recurrent parent | MO (%) | HRR (%) | ML (mm) | MB (mm) | L/B ratio | AC (%) | PC (%) | CT (min.) | ER | IR |
|---|---|---|---|---|---|---|---|---|---|---|
| BRRI31R-MASP1 | 73.90 | 72.00 | 7.00 | 2.20 | 3.18 | 22.90 | 7.90 | 17.20 | 1.30 | 4.45 |
| BRRI31R-MASP2 | 74.00 | 71.70 | 7.10 | 2.20 | 3.23 | 23.05 | 8.00 | 17.88 | 1.31 | 4.48 |
| BRRI31R-MASP3 | 74.80 | 73.00 | 7.30 | 2.29 | 3.19 | 23.38 | 8.20 | 18.10 | 1.34 | 4.50 |
| BRRI31R-MASP4 | 74.50 | 73.00 | 7.22 | 2.30 | 3.14 | 23.30 | 8.08 | 18.25 | 1.32 | 4.51 |
| BRRI31R-MASP5 | 73.70 | 71.50 | 7.00 | 2.20 | 3.18 | 23.08 | 7.80 | 17.84 | 1.29 | 4.45 |
| BRRI31R-RP | 74.00 | 72.00 | 7.20 | 2.30 | 3.13 | 23.00 | 7.90 | 18.30 | 1.30 | 4.50 |
| Mean | 74.15 | 72.20 | 7.14 | 2.25 | 3.18 | 23.12 | 7.98 | 17.93 | 1.31 | 4.48 |
| LSD (5%) | 0.44 | 0.68 | 0.13 | 0.06 | 0.04 | 0.19 | 0.15 | 0.42 | 0.02 | 0.03 |
| CV (%) | 0.56 | 0.90 | 1.73 | 2.36 | 1.14 | 0.80 | 1.81 | 2.25 | 1.37 | 0.59 |

Note. MO:Milling outturn, HRR:Head rice recovery, ML:Milled rice length in mm, MB:Milled rice breadth in mm, L/B:Length breadth ratio, AC:Amylose content in percent, PC:Protein content in percent, CT:Cooking time, ER:Elongation ratio, IR:Imbibition ration.

segregating generation through foreground and phenotypic selection. Therefore, it was proceeded by three backcrosses ($BC_3$) and advanced $BC_3F_1$ to $BC_3F_5$ generation by phenotypic selection and foreground selection to ensure the presence of the desired genes with morphological characteristics similar to recurrent parent.

In the present study, the segregated population of $BC_3F_2$ were analyzed against *xa5*, *xa13* and *Xa21* genes by chi-square ($\chi2$) test. Based on gene combinations of cross combination BRRI31R × IRBB60, it was observed that the segregated genotypic ratio was 1:2:1, which fitted expected segregation ratio for single gene model. The inheritance pattern of BB resistance controlled by single gene and segregation analysis of *xa5*, *xa13* and *Xa21* genes revealed that the chi-square ($\chi2$) value is lower than the tabulated value at 5% (5.99) and 1% (9.21) levels which prove that these genes individually followed the first law of Gregor Johan Mendel [59, 60].

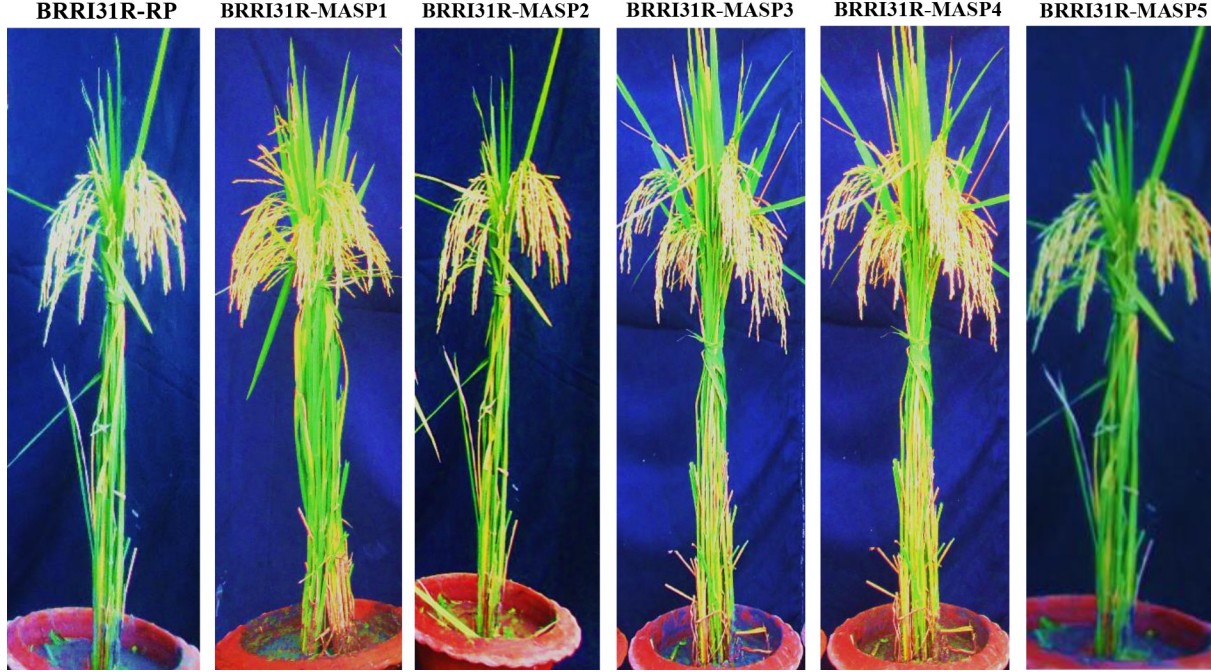

**Fig 5. Phenotype of the selected five pyramided restorer lines of $BC_3F_5$ generation compared with recurrent parent BRRI31R.**

Five homozygous pyramided restorer lines were selected from the cross combination, BRRI31R × IRBB60, with three BB resistance genes, *xa5 + xa13 + Xa21* in BC$_3$F$_5$ progenies, which showed highly resistant reaction against BB pathogens. Combination of more than one resistance genes is effective to defend BB pathogen and genes combination of *xa5 +xa13 + Xa21* are effective against the wide range of BB isolates of India [56, 61–63]. It was noted that BRRI31R had the *Xa4* gene by default and for this reason, finally, five homozygous pyramided restore lines were selected with four BB resistance genes, *Xa4 + xa5 + xa13 + Xa21* from the cross combination, BRRI31R × IRBB60 in BC$_3$F$_5$ progenies. Though *Xa4* is not effective against the BB pathogen of Bangladesh but combination of this gene might enhance the function of other resistance gene and combination of these four genes were also found as very effective against the diverse isolates of BB pathogens [23, 25, 33, 61, 64]. These gene combinations also gives long-lasting and broad-spectrum resistance in the R8012 restorer line for Chinese CMS Zhong-9A restoration [65].

All selected pyramided restorer lines with respected recurrent parent were screened against five *BXo* races. It was observed that all pyramided lines showed resistance to highly resistance to all five races of *Xoo* and mean lesion length below 3 cm and ranged from 0.2 to 1.3 cm. More than one gene, specifically a combination of *xa5+xa13+Xa21* is effective against more than one BB virulent isolates of Bangladesh [25, 33]. The data indicates that pyramided BB resistance genes in parental lines of hybrid rice could improve resistance to *Xoo*. Four genes combination (*Xa4 + xa5 + xa13 + Xa21*) showed highly resistant and comparatively showed a higher degree of resistance than parent and susceptible check and findings are consistent with previous investigations [23, 29, 32, 56, 61].

The discovery of restorers is critical for the commercialization of heterosis breeding programs in hybrid rice utilizing the cytoplasmic male sterility (CMS) method [66]. In this study, all five pyramided restorer lines viz., BRRI31R-MASP1, BRRI31R-MASP2, BRRI31R-MASP3, BRRI31R-MASP4 and BRRI31R-MASP5 showed fertility restorer alleles of *Rf3* and *Rf4*. F$_1$s pollen were studied under compound microscopic for fertility test and seed set (%) was observed under field conditions. Under both observations, all restorer lines displayed full fertility. So, all pyramided restorer lines possessed fertility restoring ability and it proves that the resistance genes of the study had not any negative effect on the restoring ability [49, 67]. These pyramided restorer lines will be used in line × tester analysis for finding the suitable combiner with the best heterotic combination compared to standard checks by General Combining Ability (GCA) and Specific Combining Ability (SCA) analysis. After molecular screening, selected five pyramided restorer lines were screened against 20 QTLs trait using SNP markers. In this study, out of 20 QTL, presence of 5 QTLs were also confirmed through gene based/linked markers.

The agronomic characters analysis revealed that BC$_3$F$_5$ restorer lines derived from the BRRI31R were more or less similar to the recurrent parent. Advanced lines of the backcross population are similar to the recipient parent because of background recovery [32]. However, out of five, two pyramided restorer lines produced higher grain yields per plant than the recurrent parent. In addition, all pyramided restorer lines were not significantly differed in terms of milling outturn, head rice recovery, milled rice length, milled rice breadth, length breadth ratio, amylose content, protein content, cooking time, elongation ratio and imbibition ratio, indicating that BC$_3$F$_5$ pyramided lines had grain quality consistent with BRRI31R. The findings revealed that there were no decreases in yield or grain quality as a result of the pyramiding of the four BB resistance genes (*Xa4, xa5, xa13 and Xa21*).

## Conclusion

We have successfully introgressed BB resistance genes into a promising restorer line, BRRI31R using MABC. The pyramided restorer lines, BRRI31R-MASP1, BRRI31R-MASP2 and

BRRI31R-MASP5 consisted of *Xa4*, *xa5*, *xa13*, *Xa21*, *Chalk5*, *Rf3* and *Rf4* genes. Moreover, BRRI31R-MASP3 and BRRI31R-MASP4 consisted of blast resistance gene *Pita* with *Xa4*, *xa5*, *xa13*, *Xa21*, *Chalk5*, *Rf3* and *Rf4* genes and these two pyramided restorer lines showed high resistant reactions against *BXo* races. It also demonstrated the utility of foreground selection in discovering target genes for bacterial blight resistance. The developed pyramided restorer lines, BRRI31R-MASP3 and BRRI31R-MASP4 with fertility restore genes could be used as donor parent to transfer bacterial blight resistance genes into susceptible restorer lines of promising hybrid rice variety. Moreover, BRRI31R-MASP3 and BRRI31R-MASP4 might be used for parental lines improvement program by (R × R) crossing and also be used as a male parent for the development of BB resistant hybrid varieties. The newly developed resistant lines of this study could be used as a replacement of restorer line of our popular hybrid rice variety BRRI hybrid dhan5 and BRRI hybrid dhan7.

## Supporting information

**S1 Table. PCR profile for gene-linked/ specific primers against Xa4, xa5 xa13 and Xa21 genes.**
(DOCX)

**S2 Table. List of trait-based SNP markers used to screen the pyramided restorer lines.**
(DOCX)

**S3 Table. List of testcross F1s of pyramided lines.**
(DOCX)

**S1 Raw images. Raw images file of the experiment.**
(DOCX)

**S1 Dataset. Raw dataset of the experiment.**
(XLSX)

## Acknowledgments

The authors are highly grateful to the Plant Pathology Division and Hybrid Rice Division for providing all necessary laboratory and field supports.

## Author Contributions

**Conceptualization:** Anowara Akter, Lutful Hassan, Md. Jamil Hasan, Mohammad Abdul Latif.

**Data curation:** Anowara Akter, Sheikh Arafat Islam Nihad, Anika Tabassum.

**Formal analysis:** Anowara Akter, Sheikh Arafat Islam Nihad.

**Funding acquisition:** Anowara Akter, Md. Jamil Hasan, Mohammad Abdul Latif.

**Investigation:** Anowara Akter, Lutful Hassan, Md. Jamil Hasan, Arif Hasan Khan Robin, Mohammad Abdul Latif.

**Methodology:** Anowara Akter, Sheikh Arafat Islam Nihad, Md. Jamil Hasan, Mahmuda Khatun, Anika Tabassum, Mohammad Abdul Latif.

**Project administration:** Anowara Akter, Md. Jamil Hasan, Mohammad Abdul Latif.

**Resources:** Anowara Akter, Sheikh Arafat Islam Nihad, Md. Jamil Hasan, Arif Hasan Khan Robin, Mahmuda Khatun, Mohammad Abdul Latif.

**Software:** Anowara Akter, Sheikh Arafat Islam Nihad.

**Supervision:** Anowara Akter, Md. Jamil Hasan, Mohammad Abdul Latif.

**Validation:** Anowara Akter, Sheikh Arafat Islam Nihad, Mohammad Abdul Latif.

**Visualization:** Anowara Akter, Sheikh Arafat Islam Nihad, Mohammad Abdul Latif.

**Writing – original draft:** Anowara Akter, Sheikh Arafat Islam Nihad, Anika Tabassum, Mohammad Abdul Latif.

**Writing – review & editing:** Anowara Akter, Sheikh Arafat Islam Nihad, Mohammad Abdul Latif.

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
