## [Decision Letter · Decision Letter 0]

4 Jan 2024

PONE-D-23-34671Pyramiding of bacterial blight resistance genes into promising restorer BRRI31R line through marker-assisted backcross breeding and evaluation of agro-morphological and physiochemical characteristics of developed resistant restorer linesPLOS ONE

Dear Dr. Latif,

Thank you for submitting your manuscript to PLOS ONE. After careful consideration, we feel that it has merit but does not fully meet PLOS ONE’s publication criteria as it currently stands. Therefore, we invite you to submit a revised version of the manuscript that addresses the points raised during the review process.

We look forward to receiving your revised manuscript.

Kind regards,

Umakanta Ngangkham, Ph. D.

Academic Editor

PLOS ONE

“Authors received funds from Strengthening Physical Infrastructure and Research Activities (SPIRA) Project, Bangladesh Rice Research Institute, through the Ministry of Agriculture, Bangladesh to conduct this experiment.”

“The authors would like to express their gratitude to the Project Director for the financial assistance provided by the Strengthening Physical Infrastructure and Research Activities (SPIRA) Project, Bangladesh Rice Research Institute, through the Ministry of Agriculture, Bangladesh. The authors are highly grateful to Plant Pathology Division and Hybrid Rice Division for providing all necessary support. The authors are extremely thankful to the Plant Pathology Division and the Hybrid Rice Division for providing all required assistance.”

“Authors received funds from Strengthening Physical Infrastructure and Research Activities (SPIRA) Project, Bangladesh Rice Research Institute, through the Ministry of Agriculture, Bangladesh to conduct this experiment.”

Additional Editor Comments:

The manuscript was well written and interesting. However, there are some minor clarification requirement from the authors.

1. What is the concentration of bacterial cell for inoculation on parent and selected pyramided restorer lines?

2. In Table 2, RP, DP, RC and Sc were mentioned. What are they?

3. The suggestions of the reviewer (2) may be incorporated in the revised manuscript.

Reviewers' comments:

Reviewer's Responses to Questions

**Comments to the Author**

1. Is the manuscript technically sound, and do the data support the conclusions?

Reviewer #1: No

Reviewer #2: Yes

2. Has the statistical analysis been performed appropriately and rigorously? 

Reviewer #1: Yes

Reviewer #2: Yes

3. Have the authors made all data underlying the findings in their manuscript fully available?

Reviewer #1: Yes

Reviewer #2: Yes

4. Is the manuscript presented in an intelligible fashion and written in standard English?

Reviewer #1: Yes

Reviewer #2: Yes

5. Review Comments to the Author

Reviewer #1: I appreciate the author who has taken the effort since It has been a burning issue in rice cultivation at this time since the Rice (Oryza sativa L.) is the world's most influential and economically significant crop.

Reviewer #2: The manuscript by Akter and colleagues focuses on the pyramiding of four bacterial blight resistance genes in the background of BRRI31R through marker-assisted backcross breeding. The manuscript is well written, and the figures are well presented. While the topic will have broad readers, are few minor revisions are required. As such, I would recommend important changes to be taken into account in the manuscript before acceptance for publication.

Comments/queries that need to be addressed:

1. The author claims that restorer line BRRI31R carries the Xa4 gene in the background based on the amplification of resistant allele corresponding to donor parent IRBB60 and resistant check IRBB4. In this regard, please consider the following points: (a) Please provide the parentage of BRRI31R if the Xa4 has been transferred from the IRBB4. (b) Kindly mention if the marker is functional and also provide the information on the response of IRBB4.

2. The author tested 200 BC3F2 with linked markers and identified homozygous plants with single, double, and triple gene combinations. Please mention if the author has checked the effectiveness and spectrum of various resistance gene combinations against five Xoo races of Bangladesh to ensure its effectiveness/durability without having negative impacts on other agronomic traits.

3. It is important to mention the initial population size for BC3F2. The 200 is a very small population to select the combination of four genes. The chances of escape of the gene are high in this population. Additionally, the selection of a superior, high-yielding, early maturing line in such a small is difficult.

4. The pollen fertility of IR79156A x BRRI31R-MASP2 and IR79156A x BRRI31R-MASP5 is 91.95 and 93.33, respectively. But why they are not considered for further analysis?

5. Why two pyramiding lines MASP3 and MASP4 are selectively biased for having improved agronomic traits over the other three pyramided lines as evidenced by Table 7 and Table 8? Similarly, for instance, Table 4 shows the mean lesion length of these two lines very similar to the donor over the other three pyramided lines and they carry the pita gene as well.

6. Line 254- A gel photograph…. It seems that the sentence is incomplete. Similarly,

7. The reference Suh et al. 2013 is repeated twice (50 and 51).

8. Check the reference 56 if it is relevant.

6. PLOS authors have the option to publish the peer review history of their article (what does this mean?). If published, this will include your full peer review and any attached files.

Reviewer #1: **Yes: **Dr. Ayyasami Ramanathan, Professor-Plant Pathology, Tamilnadu Agricultural University, Thanjavur, Tamilnadu, India

Reviewer #2: **Yes: **Dr. Kishor Kumar

---

## [Author Response · Author response to Decision Letter 0]

18 Feb 2024

Response to Editor and Reviewers comments

Comment 1. Please ensure that your manuscript meets PLOS ONE's style requirements, including those for file naming. The PLOS ONE style templates can be found at

Response: We checked and tried to arrange the manuscript according to PLOS one journal format.

Comment 2. We suggest you thoroughly copyedit your manuscript for language usage, spelling, and grammar. If you do not know anyone who can help you do this, you may wish to consider employing a professional scientific editing service.

Response: We tried to revise the language of our manuscript. The manuscript has been edited by Dr. Mohammad Abdul Latif, Director, Admin and Common Service and Former Head and Chief Scientific Officer of the Plant Pathology Division of Bangladesh Rice Research Institute (BRRI), Gazipur, Bangladesh.

Comment 3. Thank you for stating the following financial disclosure:

“Authors received funds from Strengthening Physical Infrastructure and Research Activities (SPIRA) Project, Bangladesh Rice Research Institute, through the Ministry of Agriculture, Bangladesh to conduct this experiment.”

Response: We already included the statement "The funders had no role in study design, data collection and analysis, decision to publish, or preparation of the manuscript” in the funding section.

Comment 4. Thank you for stating the following in the Acknowledgments Section of your manuscript:

“The authors would like to express their gratitude to the Project Director for the financial assistance provided by the Strengthening Physical Infrastructure and Research Activities (SPIRA) Project, Bangladesh Rice Research Institute, through the Ministry of Agriculture, Bangladesh. The authors are highly grateful to Plant Pathology Division and Hybrid Rice Division for providing all necessary support. The authors are extremely thankful to the Plant Pathology Division and the Hybrid Rice Division for providing all required assistance.”

“Authors received funds from Strengthening Physical Infrastructure and Research Activities (SPIRA) Project, Bangladesh Rice Research Institute, through the Ministry of Agriculture, Bangladesh to conduct this experiment.”

Response: We removed the funding information from the Acknowledgement section and updated the funding statement in the Funding section.

Comment 5. When completing the data availability statement of the submission form, you indicated that you will make your data available on acceptance. We strongly recommend all authors decide on a data sharing plan before acceptance, as the process can be lengthy and hold up publication timelines. Please note that, though access restrictions are acceptable now, your entire data will need to be made freely accessible if your manuscript is accepted for publication. This policy applies to all data except where public deposition would breach compliance with the protocol approved by your research ethics board. If you are unable to adhere to our open data policy, please kindly revise your statement to explain your reasoning and we will seek the editor's input on an exemption. Please be assured that, once you have provided your new statement, the assessment of your exemption will not hold up the peer review process.

Response: We attached our data as a supporting file. Please check.

Comment 6. PLOS ONE now requires that authors provide the original uncropped and unadjusted images underlying all blot or gel results reported in a submission’s figures or Supporting Information files. This policy and the journal’s other requirements for blot/gel reporting and figure preparation are described in detail at https://journals.plos.org/plosone/s/figures#loc-blot-and-gel-reporting-requirements and https://journals.plos.org/plosone/s/figures#loc-preparing-figures-from-image-files. When you submit your revised manuscript, please ensure that your figures adhere fully to these guidelines and provide the original underlying images for all blot or gel data reported in your submission. See the following link for instructions on providing the original image data: https://journals.plos.org/plosone/s/figures#loc-original-images-for-blots-and-gels.

Response: We provided the raw images as supporting information.

Comment 7. Please include captions for your Supporting Information files at the end of your manuscript, and update any in-text citations to match accordingly. Please see our Supporting Information guidelines for more information: http://journals.plos.org/plosone/s/supporting-information.

Response: Included the captions for Supporting Information files and cited in the text accordingly.

Comment 8. Please review your reference list to ensure that it is complete and correct. If you have cited papers that have been retracted, please include the rationale for doing so in the manuscript text, or remove these references and replace them with relevant current references. Any changes to the reference list should be mentioned in the rebuttal letter that accompanies your revised manuscript. If you need to cite a retracted article, indicate the article’s retracted status in the References list and also include a citation and full reference for the retraction notice.

Response: Reviewed the references list and tried to correct it according to PLOSOne Journal style.

Additional Editor Comments:

The manuscript was well written and interesting. However, there are some minor clarification requirement from the authors.

Comment 9. What is the concentration of bacterial cell for inoculation on parent and selected pyramided restorer lines?

Response: Bacterial cell concentration was 3.3 × 108 CFU/mL and the OD600 value was 1, which is equivalent to this concentration.

Comment 10. In Table 2, RP, DP, RC and Sc were mentioned. What are they?

Response: Updated in the Table 2. RP means Recipient parent; DP, Donor parent; RC, Resistant check; Sc, Susceptible check.

Comment 11. The suggestions of the reviewer (2) may be incorporated in the revised manuscript.

Reviewers' comments:

Reviewer's Responses to Questions

Comments to the Author

1. Is the manuscript technically sound, and do the data support the conclusions?

Reviewer #1: No

Reviewer #2: Yes

2. Has the statistical analysis been performed appropriately and rigorously?

Reviewer #1: Yes

Reviewer #2: Yes

3. Have the authors made all data underlying the findings in their manuscript fully available?

Reviewer #1: Yes

Reviewer #2: Yes

4. Is the manuscript presented in an intelligible fashion and written in standard English?

Reviewer #1: Yes

Reviewer #2: Yes

Review Comments to the Author

Reviewer #1: 

Comment 12. I appreciate the author who has taken the effort since It has been a burning issue in rice cultivation at this time since the Rice (Oryza sativa L.) is the world's most influential and economically significant crop.

Response: Thank you for your nice complements.

Reviewer #2: 

The manuscript by Akter and colleagues focuses on the pyramiding of four bacterial blight resistance genes in the background of BRRI31R through marker-assisted backcross breeding. The manuscript is well written, and the figures are well presented. While the topic will have broad readers, are few minor revisions are required. As such, I would recommend important changes to be taken into account in the manuscript before acceptance for publication.

Response: We are very much grateful for your valuable comments. We tried to answer your queries in the following section.

Comments/queries that need to be addressed:

Comment 13. The author claims that restorer line BRRI31R carries the Xa4 gene in the background based on the amplification of resistant allele corresponding to donor parent IRBB60 and resistant check IRBB4. In this regard, please consider the following points: (a) Please provide the parentage of BRRI31R if the Xa4 has been transferred from the IRBB4. (b) Kindly mention if the marker is functional and also provide the information on the response of IRBB4.

Response: BR7013-62-1-1R X BRRI20R is the true parentage of BRRI31R. From this cross combination, the fixed line, HRB167-19-7-7-1R, was discovered and it registered is BRRI31R. In addition, we don’t know from where Xa4 is inbuilt in the inbred lines of Bangladesh but we tested that Xa4 is not working in Bangladesh by using the monogenic lines IRRBB4 (check the Table 1 of below reference 1). For your kind concern, we used the popular functional marker to identify the Xa4 gene. Please check the following paper (2 and 3) for Xa4 marker.

1.Rashid MM, Nihad SAI, Khan MAI, et al (2021) Pathotype profiling, distribution and virulence analysis of Xanthomonas oryzae pv. oryzae causing bacterial blight disease of rice in Bangladesh. J Phytopathol 169:438–446. https://onlinelibrary.wiley.com/doi/abs/10.1111/jph.13000

2.Dossa GS, Oliva R, Maiss E, Vera Cruz C, Wydra K. High Temperature Enhances the Resistance of Cultivated African Rice, Oryza glaberrima, to Bacterial Blight. Plant Dis. 2016 Feb;100(2):380-387. doi: 10.1094/PDIS-05-15-0536-RE. Epub 2015 Dec 19. PMID: 30694136.

3.Suh JP, Jeung JU, Noh TH, Cho YC, Park SH, Park HS, Shin MS, Kim CK, Jena KK. Development of breeding lines with three pyramided resistance genes that confer broad-spectrum bacterial blight resistance and their molecular analysis in rice. Rice (N Y). 2013 Feb 8;6(1):5. doi: 10.1186/1939-8433-6-5. PMID: 24280417; PMCID: PMC4883717.

Comment 14. The author tested 200 BC3F2 with linked markers and identified homozygous plants with single, double, and triple gene combinations. Please mention if the author has checked the effectiveness and spectrum of various resistance gene combinations against five Xoo races of Bangladesh to ensure its effectiveness/durability without having negative impacts on other agronomic traits.

Response: Yes, we checked the studied resistance genes against the five Xoo races. Moreover, in separate studies, we also tested the efficacy of the studied genes against the studied race and other virulent races of Bangladesh. Please see the following references. However, we have verified the efficacy and range of different resistance gene combinations and it does not adversely affect other agronomic characteristics.

1.Rashid MM, Nihad SAI, Khan MAI, et al (2021) Pathotype profiling, distribution and virulence analysis of Xanthomonas oryzae pv. oryzae causing bacterial blight disease of rice in Bangladesh. J Phytopathol 169:438–446. https://onlinelibrary.wiley.com/doi/abs/10.1111/jph.13000

2. Anik TR, Nihad SAI, Hasan MA-I, et al (2022) Exploring of bacterial blight resistance in landraces and mining of resistant gene(s) using molecular markers and pathogenicity approach. Physiol Mol Biol Plants 2022 282 28:455–469. https://link.springer.com/article/10.1007/s12298-022-01139-x

Comment 15. It is important to mention the initial population size for BC3F2. The 200 is a very small population to select the combination of four genes. The chances of escape of the gene are high in this population. Additionally, the selection of a superior, high-yielding, early maturing line in such a small is difficult.

Response: Yes, we are aware that the 200 BC3F2 population is really small, but fortunately, we was able to obtain homozygous lines that were highly similar to the recurrent parent and carried four BB resistance genes (Xa4, Xa5, Xa13 and Xa21).

Comment 16. The pollen fertility of IR79156A x BRRI31R-MASP2 and IR79156A x BRRI31R-MASP5 is 91.95 and 93.33, respectively. But why they are not considered for further analysis?

Response: This study comes from a doctoral dissertation of Anowara Akter (first author). She is also a researcher of Hybrid Rice Division of Bangladesh Rice Research Institute. She is continuing her research. There is no Hybrid line resistance against the BB pathogen of Bangladesh. As a result, she introduced BB resistance genes into the Maintainer Line and later she will transfer these genes into the corresponding CMS line to develop BB resistant hybrid variety. So her research is still ongoing.

Comment 17. Why two pyramiding lines MASP3 and MASP4 are selectively biased for having improved agronomic traits over the other three pyramided lines as evidenced by Table 7 and Table 8? Similarly, for ins

---

## [Decision Letter · Decision Letter 1]

14 Mar 2024

Pyramiding of bacterial blight resistance genes into promising restorer BRRI31R line through marker-assisted backcross breeding and evaluation of agro-morphological and physiochemical characteristics of developed resistant restorer lines

PONE-D-23-34671R1

Dear Dr. Latif,

We’re pleased to inform you that your manuscript has been judged scientifically suitable for publication and will be formally accepted for publication once it meets all outstanding technical requirements.

Kind regards,

Umakanta Ngangkham, Ph. D.

Academic Editor

PLOS ONE

Additional Editor Comments (optional):

Reviewers' comments:

Reviewer's Responses to Questions

**Comments to the Author**

1. If the authors have adequately addressed your comments raised in a previous round of review and you feel that this manuscript is now acceptable for publication, you may indicate that here to bypass the “Comments to the Author” section, enter your conflict of interest statement in the “Confidential to Editor” section, and submit your "Accept" recommendation.

Reviewer #2: All comments have been addressed

2. Is the manuscript technically sound, and do the data support the conclusions?

Reviewer #2: Yes

3. Has the statistical analysis been performed appropriately and rigorously? 

Reviewer #2: Yes

4. Have the authors made all data underlying the findings in their manuscript fully available?

Reviewer #2: Yes

5. Is the manuscript presented in an intelligible fashion and written in standard English?

Reviewer #2: Yes

6. Review Comments to the Author

Reviewer #2: The authors have properly justified the comments which I have raised. The manuscript is well formatted. Language in the manuscript is well edited. I recommend this manuscript for publication in the PLOS One.

7. PLOS authors have the option to publish the peer review history of their article (what does this mean?). If published, this will include your full peer review and any attached files.

Reviewer #2: **Yes: **Dr. Kishor Kumar

---

## [Editor Report · Acceptance letter]

2 May 2024

PONE-D-23-34671R1 

PLOS ONE

Dear Dr. Latif, 

I'm pleased to inform you that your manuscript has been deemed suitable for publication in PLOS ONE. Congratulations! Your manuscript is now being handed over to our production team.

Kind regards, 

on behalf of

Dr. Umakanta Ngangkham 

Academic Editor

PLOS ONE